# Preventive and Curative Effects of Salicylic and Methyl Salicylic Acid Having Antifungal Potential against *Monilinia laxa* and the Development of Phenolic Response in Apple Peel

**DOI:** 10.3390/plants12081584

**Published:** 2023-04-08

**Authors:** Sasa Gacnik, Alenka Munda, Robert Veberic, Metka Hudina, Maja Mikulic-Petkovsek

**Affiliations:** 1Department of Agronomy, Biotechnical Faculty, University of Ljubljana, Jamnikarjeva 101, SI-1000 Ljubljana, Slovenia; 2Agricultural Institute of Slovenia, Hacquetova 17, SI-1000 Ljubljana, Slovenia

**Keywords:** salicylates, phenylpropanoid metabolic pathway, phenolic compounds, infection, plant protection, brown rot

## Abstract

The effects of salicylic acid (SA) and one of its better-known derivatives—methyl salicylic acid (MeSA)—on the infection of apple fruits with the fungus *Monilinia laxa*, which causes brown rot, were investigated. Since research to date has focused on preventive effects, we also focused on the curative use of SA and MeSA. Curative use of SA and MeSA slowed the progression of the infection. In contrast, preventive use was generally unsuccessful. HPLC–MS was used to analyze the content of phenolic compounds in apple peels in healthy and boundary peel tissues around lesions. The boundary tissue around the lesions of untreated infected apple peel had up to 2.2-times higher content of total analyzed phenolics (TAPs) than that in the control. Flavanols, hydroxycinnamic acids and dihydrochalcones were also higher in the boundary tissue. During the curative treatment with salicylates, the ratio of TAP content between healthy and boundary tissue was lower (SA up to 1.2-times higher and MeSA up to 1.3-times higher content of TAPs in boundary compared to those in healthy tissue) at the expense of also increasing the content in healthy tissues. The results confirm that salicylates and infection with the fungus *M. laxa* cause an increased content of phenolic compounds. Curative use of salicylates has a greater potential than preventive use in infection control.

## 1. Introduction

Global food production is under constant adjustment due to climate change, resistance of harmful organisms to chemical agents and consumers’ desire for healthy food, produced with the lowest possible use of pesticides, due to their negative impact on human health and the environment [1,2,3]. The European Commission has presented a new proposal for regulation on the sustainable use of plant protection products, which among other things requires a 50% reduction in the use of pesticides by 2030 and aims to increase the proportion of agricultural land in organic production by at least 25% [4,5,6]. This has led to a search for new approaches to protect plants from pests and environmental factors, including a search for alternative, more environmentally friendly products for plant protection [7]. One of the possibilities could be the exogenous use of salicylic acid (SA) and its derivatives.

SA is a naturally occurring phenolic compound in plants, also known as 2-hydroxybenzoic acid, and is one of the major plant hormones that plays a crucial role in plant metabolism [8,9,10,11]. It has an important influence on physiological processes in plants, such as growth, plant development and their adaptations to stress factors [10] as well as on the expression of inducible traits in the activation of plant defenses when attacked by pests [12]. Methyl salicylic acid (MeSA), a volatile derivative and an inactive precursor of SA, plays a key role in long-distance signaling from infected to uninfected tissue through the phloem [13,14]. The balance between SA and MeSA in plants is controlled by the enzymes SABP2 (SA binding protein 2), which converts the inactive form of MeSA into the active form of SA [15] and SAMT1 (SA methyltransferase 1), which catalyzes the formation of MeSA from SA [16].

It is well known that treatment with SA can affect different fruit properties, such as fruit quality. Among other things, it can improve fruit color, fruit weight, fruit firmness, vitamin C content and sugar content in strawberries [17]. SA also affects the accumulation of bioactive compounds [18,19,20] through the phenylpropanoid metabolic pathway and induction of its key enzymes, such as phenylalanine ammonia-lyase (PAL), chalcone synthase/chalcone isomerase (CHS/CHI), flavanone 3β-hydroxylase (FHT) and dihydroflavonol 4-reductase (DFR) and consequently causes changes in the accumulation of some important phenolic groups, such as flavanols, flavonols, hydroxycinnamic acids and, within these, some individual phenolic compounds, such as procyanidins, chlorogenic acid and cyanidin-3-*O*-galactoside [21,22].

Activation of the SA synthesis pathway is usually triggered by pest attack or pathogen infection but can also be stimulated by exogenous use of elicitors, and it plays an important role in SA interactions with pathogens and pests and their possible suppression [23]. Exogenous use of SA has been shown to be effective against several diseases, including anthracnose, caused by the fungus *Colletotrichum gleosporioides* [24] in mango, cherry brown rot [25], apple scab, brown leaf spot in apple [26], late blight in potato [27], gray mold in strawberry [1] and blue mold in apple [28].

Fungi and bacteria cause several postharvest diseases in fruits and vegetables during storage and transportation [29]. One of the globally widespread diseases is brown rot, caused by three main fungi species, *Monilinia laxa* (Aderhold & Ruhland) Honey, *Monilinia fructigena* (Aderhold & Ruhland) Honey and *Monilinia fructicola* (Winter) Honey. It causes most of the yield losses in the postharvest stage of peaches, nectarines, apricots, sour and sweet cherries, apples and pears [30].

After reviewing the literature, we found that some studies have attempted to evaluate the effects of SA on pathogen infection by spraying or dipping in SA solutions as a preventive application [1,24,25,31]. Only one study evaluated the effect of SA on blue mold in apple fruit caused by the fungus *Penicillium expansum* [28]. The objective of this study was to investigate the effect of preventive and curative uses of SA and methylsalicylic acid (MeSA) on infection in apples with the fungus *Monilinia laxa* and their effect on phenolic compounds’ response in apple peel. 

## 2. Results

### 2.1. Effects of Preventive and Curative Use of SA and MeSA on Brown Rot

The effects of preventive and curative uses of 2.5 mM SA and MeSA solutions on the intensity of brown rot caused by *M. laxa* were estimated. Disease incidence (%) and lesion growth rate (LGR) (mm/day) were also calculated to provide a better insight and interpretation of the results. The intensity of infection of apple fruit with the fungus *M. laxa* was measured by the diameter of the lesion (Figure 1A). This increased in all treatments over the course of the sampling dates. However, differences between sampling dates among treatments were found. With curative application of 2.5 mM MeSA, the intensity of infection was significantly lower than in control or preventive treatments on all 18 days. The effects of SA for curative use were similar for the first 14 days (up to date D4) but decreased thereafter. Preventive use of SA and MeSA was generally unsuccessful and there were no significant differences from the control.

Curative use of SA and MeSA was most successful in controlling brown rot up to 3 days after infection (D1), with infection occurring in only 6.7% of apple fruits (Figure 1B). After 7 days (D2), the incidence was already 86.7% (SA) and 73% (MeSA), and 18 days after infection, the incidence was reduced by 13.3% (SA) and 20% (MeSA) with curative application.

It is interesting to look at the growth rate of the lesions (Figure 1D), which increased in all treatments on all sampling dates, but significant differences were observed between the curative and preventive applications of the solutions (D1–D4). The sign ‘†’ in Figure 1D indicates that the statistics for the treatments could not be included due to an insufficient number of repetitions, since the curative application stopped growth in the first 3 days after infection (D1) on almost all apple fruits. LGR on fruits curatively treated with SA and MeSA remained significantly lower during the first 14 days (up to date D4) compared to those in other treatments. However, after 14 days, it approached a similar level of LGR to that in the other treatments. Similar results were obtained for the average LGR of all sampling dates combined (Figure 1C).

### 2.2. Effects of Preventive and Curative Use of SA and MeSA and Infection with M. laxa on Phenolic Composition

Thirty individual phenolic compounds were identified in the peel of the ‘Golden Delicious’ cultivar, which were divided into four major phenolic groups. Quercetin-3-*O*-rutinoside, quercetin-3-*O*-galactoside, quercetin-3-*O*-glucoside, quercetin-3-*O*-xyloside, quercetin-3-*O*-arabinopyranoside, quercetin-3-*O*-arabinofuranoside and quercetin-3-*O*-rhamnoside belonged to flavonols, which formed 44–75% of the total analyzed phenolics (TAPs) in the healthy tissue and 16–40% TAPs in the boundary tissue around the lesion. Epicatechin, catechin, four procyanidin dimers (presented as total procyanidin dimers) and six procyanidin trimers (presented as total procyanidin trimers) were analyzed among flavanols, which represented 19–47% TAPs in the healthy tissue and 51–69% TAPs in the boundary tissue. Hydroxycinnamic acids (HCAs) in apple peel represented below 10% TAP, with individual representatives: *p*-coumaric acid hexoside derivative 1, *p*-coumaric acid hexoside derivative 2, 3-*p*-coumaroylquinic acid, neochlorogenic acid, 4-*p*-coumaroylquinic acid, chlorogenic acid, cryptochlorogenic acid and caffeoylsynapoyl pentoside. Phloretin- 2-*O*-xylosyl glucoside, phloridzin and 3-hydroxyphloretin-2-xyloglucoside were identified among dihydrochalcones (DHCs), which also represented less than 10% TAP. The individual phenolic compounds in healthy and boundary tissue of apple peel are shown in the Appendix A. The results for individual phenolic compounds were generally similar to the group to which they belonged, so we focused more on the differences in the content of the phenolic groups, as shown in Figure 2. A heat map was created (Figure 3) in order to provide a more easily understandable chemical composition in healthy and boundary tissue of infected fruit treated curatively or preventively with SA and MeSA. As presented in Figure 3, in all treatments, no difference in the presence of particular phenolic compounds was found, but the contents differed between the treatments. As generally shown in Figure 3, the highest content (the lowest to highest content is from yellow to red) of individual phenolic compounds was found in SA- and MeSA-treated boundary tissue of infected fruit. The exception was quercetin derivatives, which were higher in the healthy tissue of infected and treated fruits. The response of apples to infection was strongest in the boundary tissue, with an increase in flavanols (Figure 2). In particular, the values of procyanidin derivatives and epicatechin were higher than the values in healthy tissues (Figure 3). These increased even more with treatment with salicylates. 

Quercetin-3-*O*-rutinoside and quercetin-3-*O*-arabinopyranoside had the lowest and quercetin-3-*O*-rhamnoside the highest content of all identified flavanols (Figure 3, Appendix A). The content of total flavonols (Figure 2A) increased by 45% after dipping uninfected apples in SA solution and by 59% when using MeSA compared to those in the control, although the results were not statistically significant. When apples were infected with the fungus *M. laxa* and dipped in SA and MeSA solutions, they increased even more in healthy tissues, regardless of how the two solutions were used (preventive/curative) (except for PRE_INF_SA and INF_CUR_C). The highest levels of flavonols were obtained with the preventive use of SA (PRE_INF_SA; 352.29 ± 31.76 mg/kg FW), the curative use of SA (INF_CUR_SA; 640.47 ± 42.81 mg/kg FW) and the curative use of MeSA (INF_CUR_MeSA; 750.44 ± 42.00 mg/kg FW). On the other hand, lower levels of flavonols were quantified in the boundary tissue of lesions on infected apples in comparison to that in the healthy tissue. In the boundary tissues of lesions, the highest flavonol content was achieved with the curative use of MeSA (INF_CUR_MeSA; 449.73 ± 43 mg/kg FW) and was 1.8-fold higher than that in the control (INF_CUR_C).

The results for flavanols (Figure 2B) and individual representatives catechin, epicatechin and procyanidins (Appendix A) were very different from those for flavonols, since higher contents were found in the boundary tissue of the lesions of infected apple peel. Procyanidins (Figure 3, Appendix A) accounted for the largest proportion of total flavanols and catechin for the lowest in all tissues examined. The content of flavanols in uninfected apple peel did not change with the use of SA and MeSA solutions and remained the same in healthy tissues of infected apples not treated with SA and MeSA solutions (PRE_INF_C and INF_CUR_C). Curative application of SA (INF_CUR_SA: 377.88 ± 72.89 mg/kg FW) and MeSA (INF_CUR_MeSA: 522.35 ± 84.65 mg/kg FW) in infected apple peels increased the flavanol content in healthy tissues. A strong phenolic response was observed in the boundary tissue around lesions infected with the fungus *M. laxa*, since all infected apples had higher flavanol content in the boundary tissue of the lesions than in the healthy tissue. Untreated boundary tissues of infected apples had 3.4-fold (PRE_INF_C) and 4.7-fold (INF_CUR_C) higher flavanol content than healthy tissue. Even after preventive application of SA, the difference remained high (4.4-fold higher content in the boundary tissue). Preventive use of MeSA and curative use of SA and MeSA decreased the difference in content between the healthy and boundary tissue of infected apple fruits (PRE_INF_MeSA: 2.5-fold; INF_CUR_SA: 2.7-fold; INF_CUR_MeSA: 2.1-fold higher content in boundary tissue). Preventive and curative use of SA and MeSA showed the greatest increase in flavanol content in the boundary tissues around the lesions, but no significant differences in flavanol content between the application methods (preventive or curative) or the types of solution (SA or MeSA) were found.

Hydroxycinnamic acids (Figure 2C) and dihydrochalcones (Figure 2D) were less represented groups of phenolics but even here the boundary tissues of infected apples had higher content than the healthy tissues of both infected and uninfected apples. Individual representatives of the hydroxycinnamic acids responded similarly to treatments with SA and MeSA and to infection with *M. laxa,* as did the total hydroxycinnamic acids with the exception of cryptochlorogenic acid (Appendix A). Its levels did not change significantly in infected fruit between the boundary tissue and the healthy tissue, although an increasing trend in infected fruit was observed when treated with SA and even more when treated with MeSA compared to control levels in both boundary and healthy tissue. Among all representatives, chlorogenic acid (Figure 3, Appendix A) was the most abundant. Values ranged in healthy tissue from 4.04 ± 0.45 mg/g FW in uninfected and untreated fruits to 22.3 ± 4.86 mg/kg FW in infected fruit and fruit treated with MeSA. In boundary tissue of infected apple peel, values of chlorogenic acid were up to 6.6-fold higher (INF_CUR_C) compared to that in healthy tissue. Compared to untreated healthy apple peel tissue, the content in the boundary tissue increased up to 9.4 times in curatively SA-treated fruits (ranged up to 48.7 ± 3.67 mg/kg FW in INF_CUR_SA) and up to 11.7 times in curatively MeSA-treated fruits (ranged up to 60.29 ± 2.4 mg/kg FW in INF_CUR_MeSA). With the preventive use of SA and MeSA solutions, there was no such obvious increase of chlorogenic acid in the boundary tissue compared to the healthy, untreated tissue.

The response to treatments and infection of individual dihydrochalcones were the same as for total dihydrochalcones, whereby phloretin-2-*O*-xylosylglucoside (Appendix A) was the most abundant, with a range up to 16.8 ± 3.22 mg/kg FW in healthy tissue and up to 61.11 ± 2.71 mg/kg FW in boundary tissue of infected and curatively treated fruits with MeSA. The boundary tissues of treated apples had the highest content of hydroxycinnamic acids and dihydrochalcones with both methods of using solutions (curative and preventive). The content of hydroxycinnamic acids increased slightly in healthy tissue of uninfected apples when treated with SA and MeSA, while there was no difference in the content of dihydrochalcones between treatments. In the case of dihydrochalcones, this also did not change with infection or method of application of SA and MeSA in healthy tissues, while in boundary tissues, a stronger response of dihydrochalcones to infection and treatments with SA and MeSA was observed, while content in boundary tissue were much higher than in healthy tissue. The content of dihydrochalcones in the boundary tissue around lesions of control apples was 4.3-fold higher (PRE_INF_C) and 8.1-fold higher (INF_CUR_C) than those in healthy tissues. With the use of SA, the remaining content difference was still high (PRE_IN_SA: 4.5-fold; INF_CUR_SA: 4.1-fold higher in boundary tissue) but decreased with the use of MeSA to 3.9-fold (PRE_INF_MeSA) and 2.9-fold (INF_CUR_MeSA) higher in boundary tissue than in healthy tissue of infected apple fruits. Dihydrochalcones also had the highest content in infected apples that had been treated. However, a slight difference between the ways the two solutions were used was found, since curative use resulted in higher content (INF_CUR_SA: 124.75 ± 12.66 g/kg FW; INF_CUR_MeSA: 130.10 ± 6.35 g/kg FW). 

Total analyzed phenolics (TAPs) is the sum of all individual phenolic compounds in the apple peel. Treatments with SA did not affect the content of TAPs in healthy tissues of uninfected and infected fruit except for the curative application (INF_CUR_SA: 1.218 ± 122 g/kg FW), in which the content of TAPs was significantly higher than in other SA treatments. The preventive and curative application of MeSA also significantly affected the content of TAPs in healthy tissues of infected fruits and was 1.5-fold higher (PRE_INF_MeSA) and 2.7-fold higher (INF_CUR_MeSA) than those in the control treatment (PRE_INF_C and INF_CUR_C). The content of TAPs in the boundary tissue of infected apples was also higher than in healthy tissue, highest in the curative treatment with MeSA (INF_CUR_MeSA). The levels of TAPs in boundary tissue were still significantly higher in the preventive application of SA and MeSA and in the curative application of SA than in the control. No difference in the presence of any particular phenolic compound was found in any of the samples tested, but the values differed between the treatments.

To obtain a comprehensive picture of study, two PCA analyses were performed on samples of the healthy tissue and boundary tissue around the lesion caused by the fungus *M. laxa*, as well as on the main phenolic groups (flavanols, flavonols, dihydrochalcones, hydroxycinnamic acids) and TAPs (Figure 4). A biplot was made to visualize the PCA results (Figure 4A,B). In the first PCA analysis (Figure 4A), the boundary tissue around the lesion of infected apples treated preventively and curatively with SA and MeSA was considered. The first and second components of the full data of the boundary tissue PCA model accounted for 90.1% and 8.1% of the total variance, respectively. Three groups were formed. The first consisted of control treatments of both INF treatments, with samples having a low content of phenolic compounds. The second group consisted of treatments with the preventive application of SA and MeSA (PRE_INF_SA, PRE_INF_MeSA) and the curative application of SA (INF_CUR_SA), which showed average levels of phenolic compounds. The third group was represented by the curative application of MeSA (INF_CUR_MeSA), which had the highest phenolic content compared to the other groups, especially the amount of quantified flavonols.

The second PCA analysis (Figure 4B) considered healthy tissue from both infected and uninfected apples treated preventively and curatively with SA and MeSA. The first and second components of the full data of the healthy tissue PCA model accounted for 91.8% and 6.9% of the total variance, respectively. Four groups were formed. The treatments unINF_C and INF_CUR_C were included in the first group, which had the lowest content of analyzed phenolic groups and TAPs. In healthy tissues curatively sprayed with MeSA (Group 3), the content of phenols was higher than those in groups 1 and 2, especially the values of DHC, FLA and HCA. The FLO values in group 3 were significantly lower than in group 4 (INF_CUR_MeSA).

## 3. Discussion

The effects of preventive and curative use of 2.5 mM SA and MeSA solutions on the intensity of brown rot infection caused by *M. laxa* were evaluated. Preventive use of SA and MeSA was generally unsuccessful, and there were no significant differences from the control. On the other hand, infection intensity in curatively treated apple fruit was significantly lower than that in the control or preventive treatments during the first 14 days (up to date D4) but the effect decreased thereafter. Others have also reported the inefficiency of SA used preventively to control gray mold caused by the fungus *Botrytis cinerea* [32] on peach fruit 5 days after inoculation, using only a 0.5 mM solution of SA. With the addition of the yeast antagonist *Rhodotorula glutinis,* they were able to reduce the intensity of the infection significantly. Da Rocha Neto et al. [28] compared the preventive, curative and eradicative effect of a 2.5 mM SA treatment for 2 min, whereby the eradicative addition of SA inhibited blue mold by 100%, but the 2 min SA treatment had no preventive or curative effect at 4 and 10 days post inoculation. On the other hand, Wang et al. [33] reported that a preventive application of SA suppressed disease incidence and reduced the lesion diameter of *B. cinerea* in tomato fruit exposed to a 5 mM SA solution for 15 min. We assume that, in this case, the curative use of SA and MeSA was shown to be effective because of a longer dipping time in the SA and MeSA solution. This is particularly useful when storing apples, since it slows disease development and subsequent sporulation and infection of new, uninfected apples. For more successful preventive use of SA and solutions, based on the results of Wang et al. [33], it would be necessary to test higher concentrations of solutions, but not higher than 3.5 mM for this cultivar. Preliminary tests have shown that a concentration of 3.5 mM has phytotoxic effects on the cultivar ‘Golden Delicious’ [31]. 

Persistence of SA and MeSA in tissues could also be a reason why the curative use of SA and MeSA is more successful in slowing the progression of *M. laxa* infection. Detailed hourly data on the absorption of SA and MeSA for apple leaves [22] have shown that the contents of SA and MeSA after treatment are very high in the first hour, and after 3 h, the MeSA content decreases a little, while the content of SA is halved. The content of MeSA still remained quite high 6 and 9 h after treatment, while the SA content decreased 9-fold and even more in comparison with the content after 1 h. In both cases, the contents after 24 h were very low compared to the contents after 1 h. This probably means that the contents after 24 h, when the apples were infected in the preventive application, were too low for protection. In contrast, the curative treatment was applied one day after infection, suggesting that the treatment had a direct antimicrobial effect on the fungus *M. laxa*, which may be associated with disruption of the major pathway of cellular respiration [34]. SA may also bind specifically to proteins that degrade intracellular H_2_O_2_ [35], leading to an accumulation of this compound in the cell and inactivation of *Aspergillus brasiliensis* conidia [36]. Da Rocha Neto et al. [37] reported that SA also has direct antimicrobial activity against conidia of *Penicillium expansum*, probably by penetrating the cell wall and triggering numerous interactions with the plasma membrane, resulting in lipid bilayer disruption and/or damaging proteins responsible for cell permeability, thereby increasing the concentration of reactive oxygen species.

Phenolic compounds play an important role in the development of plant resistance to certain pathogens; they may be in the form of phytoanticipins, which are constantly present in plants and whose content usually increases sharply when attacked by pathogens, or in the form of phytoalexins, which are sensitized from precursors in response to attack by a pathogen, or in the form of inactive bound forms that are readily converted to a biologically active free compound when attacked by a pathogen and are much more toxic [38,39]. In our study, an analysis of phenolic compounds in apple peel under the influence of treatments with SA and MeSA and infection with *M. laxa* was performed. Thirty different phenolic compounds were identified, belonging to four different phenolic groups: flavanols, flavonols, hydroxycinnamic acids and dihydrochalcones. In all analyzed samples, no difference in the presence of various phenolic compounds between infected and uninfected fruits was found, which means that our method of analyzing the content of phenolic compounds did not detect the presence of phytoalexins but only of phytoanticipins, which in quite a few cases increased with infection in the boundary tissue around the lesion, with the exception of flavonols. In infected apples, higher levels of flavonols were quantified in healthy tissue than in boundary tissue around lesions. Various authors have reported different effects of infection on the content of flavonols in different parts of infected fruits. Mikulic Petkovsek et al. [40] reported a higher content of quercetin derivatives (major representatives of flavonols) in the healthy tissue of apple leaves of the cultivar ‘Golden Delicious’ infected with apple scab, compared to spot and boundary tissue in 2009; while in 2008, there were almost no significant differences. Similarly, a higher content of total flavonols was measured in healthy grape berries than in symptomatic ones infected with Bois noir phytoplasma [41]. In contrast, tissue infected with various viruses or pathogens was found to have more flavonols compared to healthy tissue [42,43,44,45]. The conclusion could be that the response of flavonols, which undoubtedly play an important role in plant defense mechanisms [46] to pathogen infection depends on various factors, such as fungus species and different growth season, because in some cases a higher content of flavonols accumulates in the boundary tissue [42,44,45], which can act as a kind of barrier and stop the growth and development of the fungus [47], although elsewhere the content is higher in the healthy tissue [40] and thus a larger part of the fruit could shift to protection from the pathogen. The fact that the healthy tissue had a higher content of flavonols than the boundary tissue around the lesion caused by the fungus *M. laxa* could be crucially influenced by the treatment with salicylates, since in the case of the curative use of the solutions, a significant difference between the treated and untreated fruits was revealed. It is already known that salicylates increase the content of flavonols in strawberries [17], apple fruit [21], apple leaves [22] and grape berries [48]. The results for preventive use of solutions are less clear, with the exception of treatment with MeSA. 

The results for flavanols, hydroxycinnamic acids and dihydrochalcones were reversed, since in all cases, a higher content in the boundary tissue was recorded compared to the healthy. Others have also reported similar findings for the mentioned phenolic groups [40,44,45]. A strong phenolic response with flavanols and dihydrochalcones was observed in the boundary tissue around lesions infected with the fungus *M. laxa*, since all infected apples had much higher content in the boundary tissues around the lesions than did the healthy tissues. Untreated boundary tissues of infected apples had a 3.4-fold (preventive) and 4.7-fold (curative) higher content of flavanols than healthy tissues. 

Individual representatives of the hydroxycinnamic acids in general responded similarly to treatments with SA or MeSA and to infection with *M. laxa*, as total hydroxycinnamic acids. Chlorogenic acid, the most abundant representative, was in boundary tissue of infected apple peel up to 6.6-fold higher (INF_CUR_C) compared to healthy tissue. Increasing contents of chlorogenic acid and its derivatives have already been associated with a reduction in the susceptibility of fruit to brown rot infections [49,50,51,52]. Villarino et al. [50] reported that chlorogenic acid concentrations, similar to those in peach fruit, does not inhibit spore germination or mycelial growth of fungus *M. laxa* in culture but inhibits the production of melanin-like pigments in the mycelia of *M. laxa* in culture (42% melanin reduction). Lee and Bostock [49] also reported that chlorogenic acid has antifungal potential by inhibition of the production of the cell wall degrading enzymes polygalacturonase and cutinase in *M. fructicola* cultures and inhibition of appressorium formation from germinated conidia and subsequent brown rot lesion development. 

Compared to untreated healthy apple peel tissue, the content in the boundary tissue increased up to 9.4 times in curatively SA-treated fruits (INF_CUR_SA) and up to 11.7 times in curatively MeSA-treated fruits (INF_CUR_MeSA). With the preventive use of SA and MeSA solutions, there was no such obvious increase of chlorogenic acid in the boundary tissue compared to that in the healthy, untreated tissue. Curative treatment of fruit with SA and MeSA significantly increased chlorogenic acid levels compared to untreated fruit, which could be one of the main reasons why curative use of SA and MeSA was more successful in slowing the progression of *M. laxa* infection.

At the expense of increasing the content of TAPs in healthy tissue of treated apples, the difference in content between healthy and boundary tissue of infected apple fruits decreased, especially with the curative use of SA (1.2-times higher TAP content in boundary tissue) and MeSA (1.4-times higher TAP content in boundary tissue). TAPs also increased significantly in boundary tissue of infected fruits, especially in the treatment with SA and MeSA, regardless of the manner of use. It has been suggested that SA inhibits catalase activity, which leads to increased H_2_O_2_ levels, which is considered to be one of the main co-compounds formed during oxidative stress in plants [35] and induces PAL gene expression [53] and synthesis of phenolic compounds [54]. However, it is interesting to note that the content of TAPs is greatly increased even in healthy tissue when treated curatively with SA and MeSA and preventively with MeSA. Healthy tissue treated with MeSA may have had such high content even with preventive application because of the absorption of the compounds into the apple tissue and better membrane permeability of MeSA. Gačnik et al. [22] reported that the absorption of MeSA in apple peel of the cultivar ‘Topaz’ was significantly better than the absorption of SA. The content of SA in the apple peel of the treated fruit was around 0.5 µg/g DW up to 6 h after treatment and then started to decrease, while the content of MeSA in treated fruits was almost 6-fold higher than that of SA in the first 6 h. Bearing in mind that the fruits were infected 24 h after the treatments in the application, it is possible that the content of SA in the apple peel was too low for protection. The MeSA content was almost halved within 24 h, but still remained about 5-fold higher than the content of SA [22]. In addition, MeSA is considered to be a volatile molecule, which is effectively transported over long distances in plants [55]. However, we cannot say this with certainty because there was no increase in uninfected fruits treated with MeSA. 

## 4. Materials and Methods

### 4.1. Plant Material, Pathogen and Preparation of SA and MeSA Solutions

This study was performed on the fruit of the standard apple cultivar ‘Golden Delicious’, grown at the Biotechnical Faculty in Ljubljana and stored in a cold chamber after harvest. Graphical scheme of the experimental design is presented in the Appendix A. Before use, apples were disinfected with NaClO solution (1% active chlorine) for 2 min, rinsed in distilled water and air-dried. To prepare the spore suspension of *M. laxa*, previously infected fruits (infection with agar pieces covered with the mycelium of *M. laxa*) were used with abundant sporulation of the fungus on the surface of the fruit. The spores were scraped from the surface of the fruit, suspended in sterile distilled water (SDW) and adjusted to the desired concentration.

SA and MeSA were obtained from Sigma-Aldrich (St. Louis, MO, USA), with Sigma product codes 247588 (SA) and 167037 (MeSA), and diluted in SDW using a magnetic bar and stirrer to a concentration of 2.5 mM. The SA solution was prepared by dissolving 3.453 g of SA in a 0.5 L beaker with SDW and the MeSA solution by dissolving 3.804 g of MeSA in a 0.5 L beaker with SDW with constant stirring. Both solutions were then poured into a 10 L canister for water, to which 1% DMSO (dimethyl sulfoxidine) and 0.1% Tween 20 were added. Then, SDW was added up to the 10 L mark and mixed well.

### 4.2. Treatments, Infection and Sampling

Disinfected apples were distributed in plastic boxes on wet cardboard bottoms and divided among nine treatments: unINF_C (disinfected apples, immersed in distilled water and uninfected), unINF_SA (disinfected apples, immersed in SA and uninfected), unINF_MeSA (disinfected apples, immersed in MeSA and uninfected), PRE_INF_C (disinfected apples, immersed in distilled water and infected), PRE_INF_SA (disinfected apples, immersed in SA and infected), PRE_INF_MeSA (disinfected apples, immersed in MeSA and infected), INF_CUR_C (disinfected apples, infected and immersed in distilled water after 24 h), INF_CUR_SA (disinfected apples, infected and immersed in SA after 24 h) and INF_CUR_MeSA (disinfected apples, infected and immersed in MeSA after 24 h). There were three replicates per treatment (three plastic boxes), and each plastic box contained 15 apples.

To test the preventive application, apples were dipped in a 2.5 mM solution of SA and MeSA for half an hour and then air dried for 2 h. Apples from all INF treatments were then infected through a puncture wound in the equatorial region with a standardized needle (3 mm deep × 1 mm wide), through which 20 µL of a suspension of spores of the fungus *M. laxa* was applied with a concentration of 5 × 10^5^ spores/mL. Controls were unwounded and wounded by application of distilled water. Apples from all treatments were then incubated for 24 h in a growth chamber at a controlled temperature (20 °C) and 100% humidity. After 24 h of infection, apples destined for cure were removed from the growth chamber and immersed in a 2.5 mM solution of SA and MeSA for half an hour. After drying, they were returned to the growth chamber, and apples from all treatments were incubated for 18 days under controlled conditions at 17 °C and 100% humidity.

First, the intensity of infection (measurement of the diameter of lesions caused by *M. laxa*) was assessed at 3 days (D1), then at 7 days (D2), 10 days (D3), 14 days (D4) and 18 days (D5). Based on the values of lesion diameters, the lesion growth rate (LGR) was estimated as follows: LGR = (Σ(θt)/t), where θ represents the average diameter of the lesion at time “t” for each sampling date (D1–D5). Average LGR was calculated based on all five sampling dates. Results were expressed in mm/day. Disease incidence was calculated by the division of the number of successfully infected apples presenting with characteristic symptoms of brown rot by the total number of apples. The average results were expressed in %.

On the sampling date (D5) of the measurements, healthy tissue and a marginal section of the lesion with a 1–2 mm narrow strip of healthy tissue (boundary tissue) were removed from the apples and immediately shock-frozen in liquid nitrogen and stored at −20 °C for analysis of phenolic compound content by HPLC–MS systems.

### 4.3. Individual Phenolic Compounds HPLC–MS Analysis of Apple Peel

For analysis of phenolic compounds in healthy and boundary tissue of apple peel, 2 g of apple peel (ground with liquid nitrogen) was weighed and extracted with 7 mL of a solution of 80% methanol from Sigma-Aldrich and 3% formic acid from Fluka Chemie. Phenolic compounds were extracted in a cooled ultrasonic bath. After 45 min of extraction, samples were centrifuged using an Eppendorf Centrifuge 5810 R for 5 min at 10,000 rpm at 4 °C and were filtered into vials using a 0.20 µm Chromafil AO-20/25 polyamide (Macherey-Nagel) filter.

Quantification of individual phenolic analysis was performed with high-performance liquid chromatography (HPLC; Thermo Scientific; San Jose, CA, USA) as described by Gacnik et al. (2021b). Identification of phenolic compounds was performed with an LTQ XLTM Linear Ion Trap Mass Spectrometer (Thermo Scientific) with electrospray ionization (scanning from m/z 115–1500; operating in negative mode for identification of all phenolic groups) and diode array detector (DAD) set on two different wavelengths: 280 and 350 nm. A Gemini C18 (150 × 4.6 mm, 3 μm) from Phenomenex (Torrance, USA) was used as the column, and it was set at 25 °C. Mobile phases for combined analysis of phenolic compounds by HPLC–MS consisted of a mixture of mobile phases A and B (A: 97% acetonitrile + 0.1% formic acid; B: 3% acetonitrile + 0.1% formic acid). 

For calculation of the content of each phenolic compound in healthy and boundary tissues of apple peels, different standards were used, as represented in Table 1. Calculations were performed from calibration curves of external standards and peak areas of the corresponding phenolic compound. Results were expressed in mg/kg FW of apple peel.

### 4.4. Statistical Analysis

R-commander (R Formation for Statistical Computing, Auckland, New Zealand) statistical software was used to analyze the data. The effects of preventive and curative use of the SA and MeSA treatments on infection intensity, disease incidence and phenolic content of major groups and individual phenolic compounds were subjected to one-way analysis of variance (ANOVA). Significant differences among treatments were calculated by multiple comparisons of means using the Tukey test (*p* < 0.05).

Principal component analysis (PCA) was performed for the content of the main phenolic groups (flavanols, flavonols, dihydrochalcones and hydroxycinnamic acids) and TAPs (total analyzed phenolics) in healthy and boundary tissue around lesions caused by the fungus *M. laxa* (Figure 4). 

## 5. Conclusions

Curative use of 2.5 mM solutions SA and MeSA slowed the progression of brown rot infection caused by the *M. laxa* fungus. On the other hand, preventive use was generally unsuccessful and there were no significant differences with the control. This may be because of the low persistence of SA and MeSA in apple peel after treatment. When analyzing the content of phenolic compounds by HPLC–MS, no phytoalexins were detected but only phytoanticipins, which in some cases increased with infection in the boundary tissue around the peel, except for flavonols, whose content was higher in the healthy tissue of the infected fruits. On the other hand, flavanols, hydroxycinnamic acids and dihydrochalcones were higher in the boundary tissue of the lesion, and the difference in the content of compounds between healthy and marginal tissue was greater. We hypothesize that dihydrochalcones and flavanols elicited a stronger phenolic response. The use of salicylates decisively increased the content of flavanols, hydroxycinnamic acids and dihydrochalcones in the boundary tissue around the lesion, whereas in healthy peel tissue, only curative treatments affected the content of flavonols, flavanols, hydroxycinnamic acids and total phenolic content in infected fruit. 

## Figures and Tables

**Figure 1 plants-12-01584-f001:**
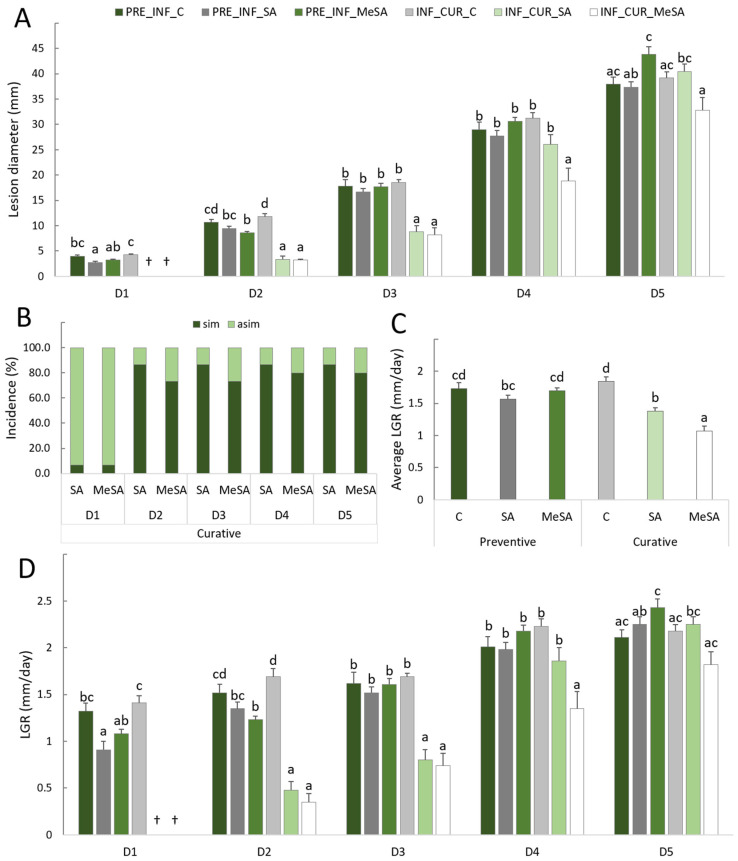
Curative (cur) and preventive (pre) treatments of apple fruits with 2.5 mM SA and MeSA solutions and infection with *M. laxa* (**A**) severity (mm) on different sampling dates after infection (after 3 days (D1), 7 days (D2), 10 days (D3), 14 days (D4) and 18 days (D5)), (**B**) incidence (%) in symptomatic (sim) and asymptomatic (asim) fruit, (**C**) average lesion growth rate (LGR) in 18 days (mm/day) and (**D**) LGR on individual sampling dates (mm/day). Data represent the average ± standard error. Different lower letters indicate significant differences between the treatments (Tukey, *p* ≤ 0.05).

**Figure 2 plants-12-01584-f002:**
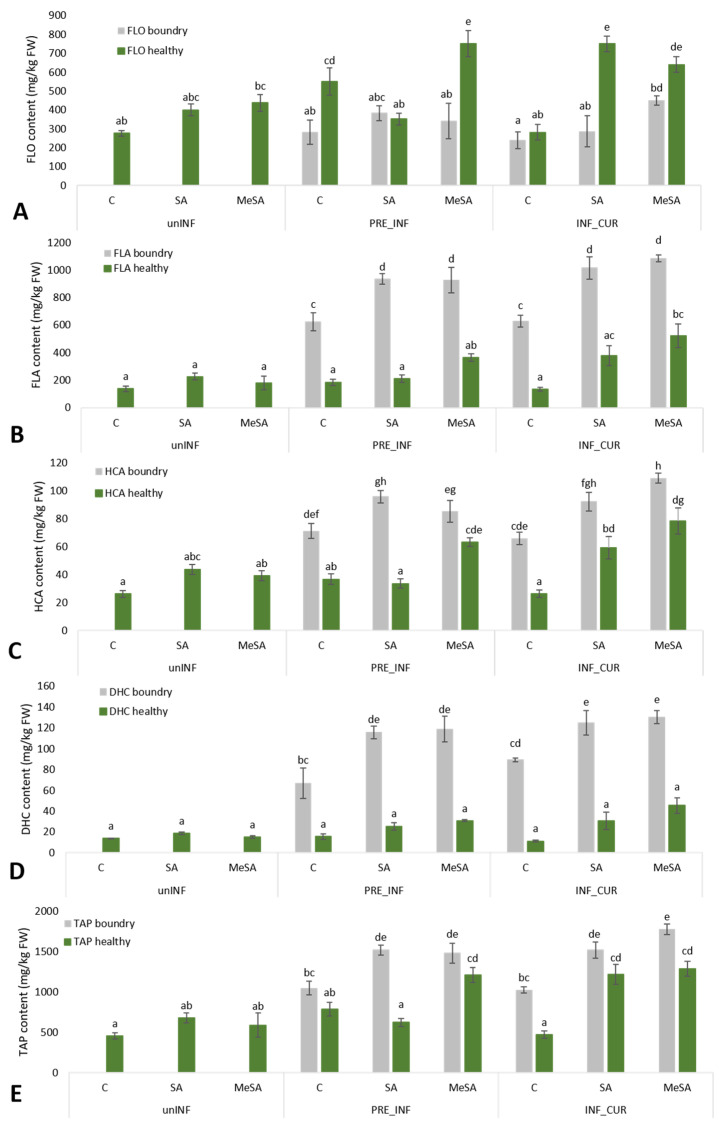
Flavonols (**A**), flavanols (**B**), hydroxycinnamic acids (**C**), dihydrochalcones (**D**) and total analyzed phenolic (TAP) (**E**) content in uninfected healthy tissue of apple peel treated with salicylic acid (SA) and methyl salicylic acid (MeSA) and in *M. laxa* infected healthy and boundary tissue around the lesion, preventively or curatively treated with SA and MeSA. Data are means ± standard error. Different letters indicate significant difference between treatments (*p* < 0.05; Tukey tests).

**Figure 3 plants-12-01584-f003:**
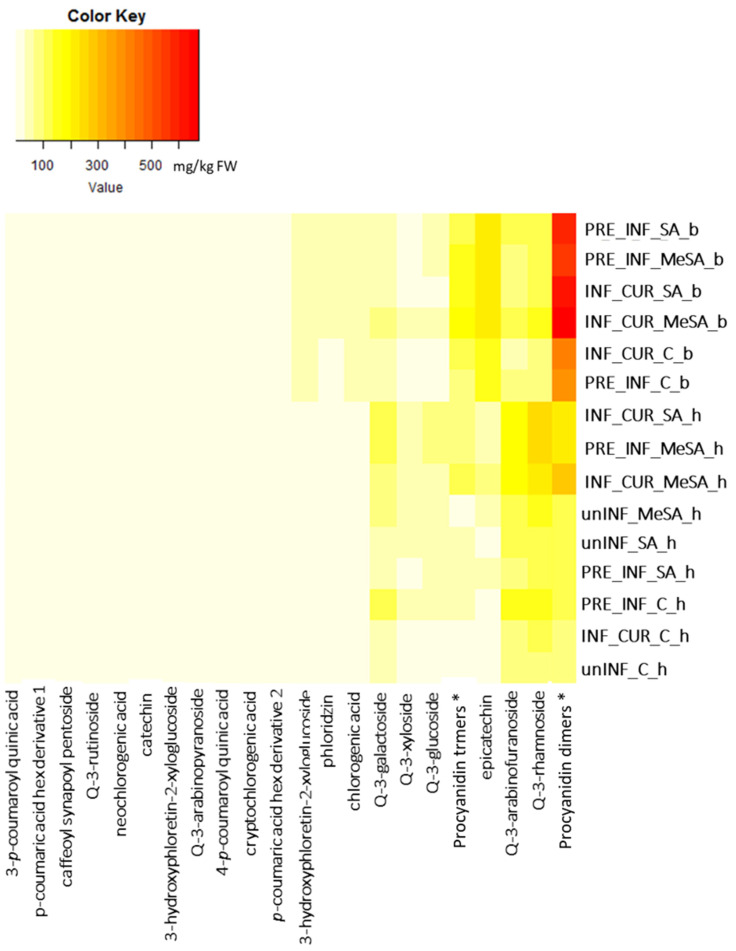
Heatmap for the content of all analyzed individual phenolic compounds (mg/kg FW) in uninfected (unINF) and infected (INF) healthy (h) and boundary (b) tissue of apple peel around the lesion caused by the fungus *M. laxa*, treated preventively (PRE) and curatively (CUR) with salicylic acid (SA) and methyl salicylic acid (MeSA). C: control, Q: quercetin, *: sum of all derivatives. Yellow, lower content of individual phenolics; red, higher content of individual phenolics.

**Figure 4 plants-12-01584-f004:**
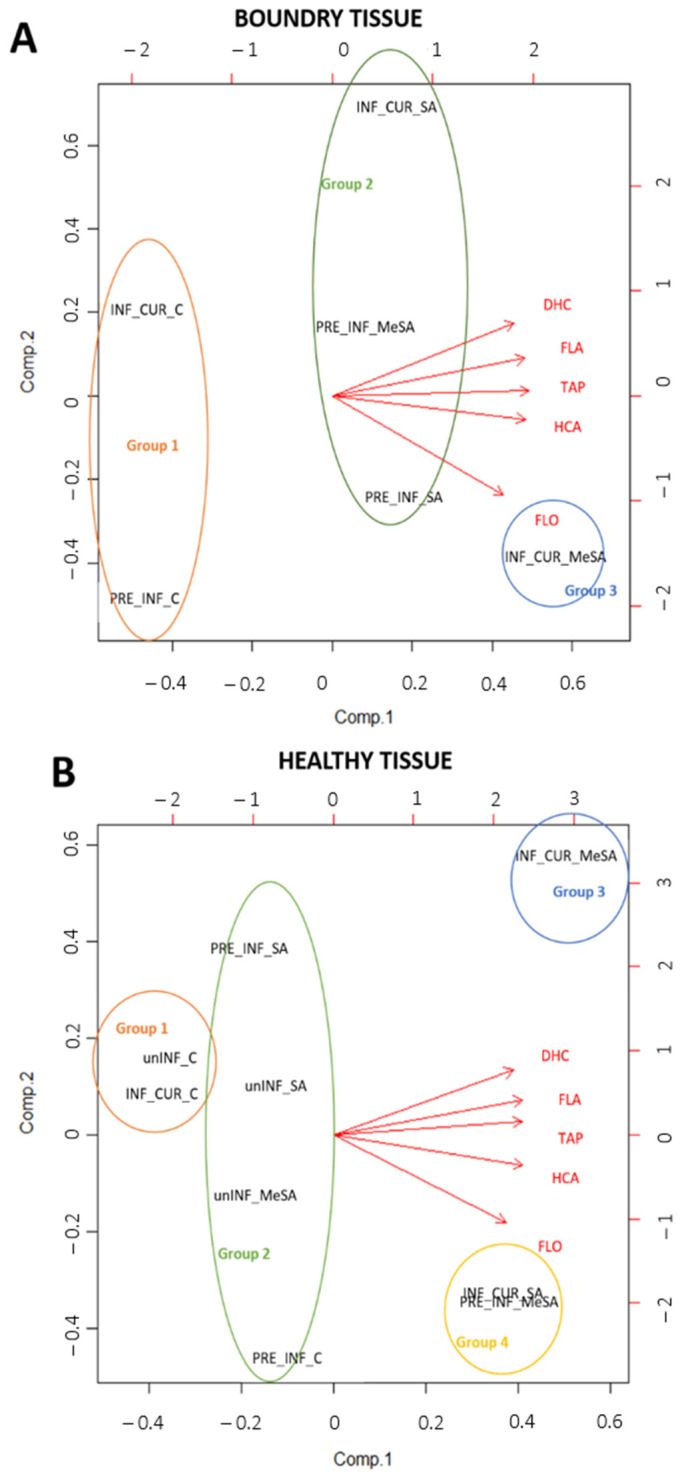
Biplot corresponding to PCA for apple peel boundary tissue around lesion caused by *M. laxa* (**A**) and healthy tissue (**B**), treated preventively (pre) or curatively (cur) with salicylic acid (SA) and methyl salicylic acid (MeSA). INF, infected; unINF, uninfected; FLO, flavonols; FLA, flavanols; HCA, hydroxycinnamic acids; DHC, dihydrochalcones; TAP, total analyzed phenolics. Groups 1 to 4: grouping of samples based on multivariate analysis.

**Table 1 plants-12-01584-t001:** Standards used for quantification of phenolic compounds.

	Used Standards	Obtained by
Flavanols	procyanidin B1	Fluka Chemie
	catechin	Fluka Chemie
	epicatechin	Fluka Chemie
Hydroxycinnamic acids	p-coumaric acid	Fluka Chemie
	sinapic acid	Fluka Chemie
	chlorogenic acid	Sigma-Aldrich
	4-caffeoylquinic acid	Sigma-Aldrich
Flavonols	quercetin-3-rutinoside	Fluka Chemie
	quercetin-3-galactoside	Fluka Chemie
	quercetin-3-glucoside	Fluka Chemie
	quercetin-3-xyloside	Fluka Chemie
	kaempferol-3-glucoside	Fluka Chemie
	quercetin-3-arabinopyranoside	Apin Chemicals LTD
	quercetin-3-arabinofuranoside	Apin Chemicals LTD
	isorhamnetin-3-glucoside	Extrasynthèse
	quercetin-3-rhamnoside	Sigma-Aldrich
Didydrochalcones	phloretin	Fluka Chemie
	phloridzin	Fluka Chemie

## Data Availability

Data are available from the corresponding author.

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
