# Peer review of "Preventive and Curative Effects of Salicylic and Methyl Salicylic Acid Having Antifungal Potential against Monilinia laxa and the Development of Phenolic Response in Apple Peel"

_plants, 2023, doi:10.3390/plants12081584_

Round 1

Reviewer 1 Report

This paper continues the studies carried out by the same group on the effects of SA and MeSA on apple crop against brown rot. The paper is well presented with results that complement previous work.

Comments to the text are given below:

Key words: indicate others that are not already included in the title of the article.

L 72-74 references?

L 84 Figure S1

L 88 indicate abbreviation SDW

L 91 replace with SDW

L 94-104 it would help the reader if the abbreviations of the specimen types were indicated in Figure S1.

L 147-155 modify the paragraph to better follow the text perhaps a table would help 

L 170-173 Phrase for the results section

L 266 modify by INF_CUR_C

L 290-294 the results of chlorogenic acid can be discussed with other references on its role in brown rot disease.

L 351 modify by INF_CUR_SA

L 352 indicate abbreviation (INF_CUR_MeSA)

Fig 4 If possible improve the size of the abbreviations for better visualization of the figure. There are errors in the abbreviations in the figure such as KUR.

L 369 replace brown mold with brown rot

Author Response

Dear Reviewer 1!

Thank you for your review of our article. We corrected what you requested. I must also remind you that table 1 has been added and that in two pictures (1 and 2) the color schemes of the charts have been changed (layout and everything else remained the same).

With kind regards,

Saša Gačnik

Reviewer 2 Report

This manuscript was well-wrote, and the experiment were well-designed, the results were interesting and valuable. And there were three little comments  for the authors to consider and revise before it was acceptted, as follow:

1. For the MeSA, authors should do some explain or provide some introduction about it and also provide something about the relationship between SA and MeSA.

2. How to determine the SA and MeSA concentrations in M & M sections in line 92, authors should provide some details or related reference.

3.  Consider removing any personal expressions through the whole manuscript - like our, we etc. 

Author Response

Dear Reviewer 2!

Thank you for your review of our article. We corrected what you requested. I must also remind you that table 1 has been added and that in two pictures (1 and 2) the color schemes of the charts have been changed (layout and everything else remained the same).

With kind regards,

Saša Gačnik

Reviewer 3 Report

Interesting and well-described study. Response is misspelled in the title.

Author Response

Dear Reviewer 3!

Thank you for your review of our article. We corrected what you requested. I must also remind you that table 1 has been added and that in two pictures (1 and 2) the color schemes of the charts have been changed (layout and everything else remained the same).

With kind regards,

Saša Gačnik
